# Ubiquitin Specific Protease USP48 Destabilizes NF-κB/p65 in Retinal Pigment Epithelium Cells

**DOI:** 10.3390/ijms23179682

**Published:** 2022-08-26

**Authors:** Serena Mirra, Laura Sánchez-Bellver, Carmela Casale, Alessandra Pescatore, Gemma Marfany

**Affiliations:** 1Department of Genetics, Microbiology and Statistics, Universitat de Barcelona, Avda. Diagonal 643, 08028 Barcelona, Spain; 2CIBERER, Instituto de Salud Carlos III, 28029 Madrid, Spain; 3Institut de Biomedicina-Institut de Recerca Sant Joan de Déu (IBUB-IRSJD), Universitat de Barcelona, 08028 Barcelona, Spain; 4Institute of Genetics and Biophysics, ‘Adriano Buzzati-Traverso’ (CNR), Via Pietro Castellino 111, 80131 Naples, Italy

**Keywords:** USP48, retina, NF-κB pathway

## Abstract

Activation of NF-κB transcription factor is strictly regulated to accurately direct cellular processes including inflammation, immunity, and cell survival. In the retina, the modulation of the NF-κB pathway is essential to prevent excessive inflammatory responses, which plays a pivotal role in many retinal neurodegenerative diseases, such as age-related macular degeneration (AMD), diabetic retinopathy (DR), and inherited retinal dystrophies (IRDs). A critical cytokine mediating inflammatory responses in retinal cells is tumor necrosis factor-alpha (TNFα), leading to the activation of several transductional pathways, including NF-κB. However, the multiple factors orchestrating the appropriate regulation of NF-κB in retinal cells still remain unclear. The present study explores how the ubiquitin-specific protease 48 (USP48) downregulation impacts the stability and transcriptional activity of NF-κB/p65 in retinal pigment epithelium (RPE), at both basal conditions and following TNFα stimulation. We described that USP48 downregulation stabilizes p65. Notably, the accumulation of p65 is mainly detectable in the nuclear compartment and it is accompanied by an increased NF-κB transcriptional activity. These results delineate a novel role of USP48 in negatively regulating NF-κB in retinal cells, providing new opportunities for therapeutic intervention in retinal pathologies.

## 1. Introduction

IRDs are a group of rare retinal disorders that cause photoreceptor dysfunction and degeneration, which results in a progressive loss of vision. Although the prevalence, etiology, pathogenesis, and clinical characteristics of IRDs are very different, common hallmarks can be identified, and include neuroinflammation, microglia activation, and mitochondrial dysfunction [1]. IRDs progression is influenced by chronic and low-grade inflammation since the release of pro-inflammatory cytokines may exacerbate photoreceptor degeneration [2]. Importantly, inflammation plays a crucial role also in the etiology and pathogenesis of retinal neurodegenerative disorders with a high prevalence, such as AMD [3], and DR [4]. Therefore, the scavenging of pro-inflammatory cytokines such as TNF-α, IL-1β, and IL-6, has been a valid therapeutic concept in these pathologies [5].

Nuclear factor kappa-light-chain-enhancer of activated B cells (NF-κB) is a widely expressed transcription factor involved in immune response as well as in the regulation of the expression of many genes related to cell survival, proliferation, and differentiation [6]. The five members of the NF-κB family (p65, RelB, c-Rel, p50, and p52) are typically sequestered in the cytosol in inactive complexes with the inhibitory IκB proteins, being IκBα the most prevalent and best studied [7]. Through combinatorial associations, the NF-κB protein family members can form up to 15 different dimers [8]. The p50/65 heterodimer clearly represents the most abundant of NF-κB dimers and it is found in almost all cell types. Most noxious cell stimuli lead to the activation of an IKK complex (IKKα, IKKβ, and NEMO), which in turn phosphorylates IκBα thus leading to its proteasomal degradation and liberating NF-κB, which translocates to the nucleus and activates its target genes [9]. Early responding transcripts encode some cytokines as well as regulators of the NF-κB pathway itself, in a regulatory loop. The expression of some early response genes is also constitutive and increases with NF-κB abundance to maintain pathway homeostasis, as is the case of *NFKBIA*, which encodes IκBα. In contrast, other genes are strictly inducible and not expressed in untreated cells (e.g., *IL6* and *IL8*) [10].

In the neural retina, TNFα is an important mediator of the innate immune response. TNFα can activate various signaling cascades, including NF-κB, and the potentially harmful role of this pathway has previously been reported for several retinal neurodegenerative conditions, where NF-κB activation may result in either cell survival or death depending on the cell type [10]. Notably, TNFα can also affect retinal pigmented epithelium (RPE) function, which is essential for photoreceptors’ viability [3].

An increasing number of studies report novel molecular actors regulating NF-κB and its function in many tissues. The ubiquitin-specific protease 48 (USP48) is expressed in almost all human tissues and is involved in several cellular processes, such as the control of the immune response, DNA repair [11], regulation of the sonic hedgehog pathway [12], and tumorigenesis [13,14,15]. USP48 displays an N-terminal ubiquitin C-terminal hydrolase (UCH) domain, required for catalytic activity [16], followed by three ubiquitin-specific protease (DUSP) domains, involved in protein-protein interaction. It also contains a regulatory CK2 domain—where three serines can be phosphorylated by the CK2 kinase, thus enhancing USP48 deubiquitinating function–as well as a ubiquitin-like (UBL) domain, which also contains a nuclear localization signal (NLS) [16,17]. Compared to other deubiquitinating enzymes, USP48 activity is rather specific, since it does not display a strong ubiquitin peptidase activity and it functions as a trimming enzyme of long K48 ubiquitin chains rather than completely disassembling ubiquitin from substrates. This trimming activity selectively rescues target nuclear proteins from proteasomal degradation and depends on the regulatory interactors in response to stimulation [17]. Reported USP48 substrates include histone H2A, Gli1, Aurora B and p65 [14,17,18,19,20].

Induced activation of p65 is tightly controlled by a broad spectrum of molecular mechanisms driving postinduction termination. p65 activity is predominantly controlled by association with IκBα, which is also a transcriptional target of NF-κB [19]. The newly synthesized nuclear IκBα acts both at the nuclear level, by removing the NF-κB/p65 complex from the DNA and terminating its transcriptional activity, and at the cytosolic level, by transporting p65 back to the cytoplasm. In the nuclei, NF-κB transcriptional activity is further modulated by the crosstalk with other transcription factors resulting in either synergistic (e.g., AP-1) or antagonistic (e.g., cEBP/β) effects [8,20]. Moreover, the differential binding kinetics of p65 with different importin/exportin proteins and the modulation by post-translational modifications provide further mechanisms of NF-κB activity regulation, overall determining the extent of nuclear translocation of p65 [21,22,23,24]. In fact, a variety of post-translational modifications, such as phosphorylation, degradative and regulatory ubiquitination, and acetylation, modulate the fine-tuning of the NF-κB/p65 activation outcomes [25]. Degradative ubiquitination plays an important role in both cytosolic and nuclear compartments, where a pool of p65 is associated with chromatin. The CRL2 SOCS1/ECS SOCS1 complex targets the chromatin-bound p65 for degradation [26,27]. However, NF-κB-dependent transcription activity could be sustained by the eight subunit (CSN1-8) CSN complex, which enhances the nuclear USP48 deubiquitinating activity. As p65 is also a direct interactor and target of USP48 [17], the CSN complex promotes USP48-mediated deubiquitination of p65 and its stabilization on chromatin. CSN also regulates p65 stability in the cytosol by modulating the IκBα-p65 interaction throughout complex mechanisms involving different USP proteins [28].

So far, the research on the contribution of USP48 in regulating p65 stability and activity has been performed in non-retinal replicative or transformed cells, such as HeLa [16,17]. However, USP48 function in the retina and RPE remains unknown. In this study, we aim to elucidate the mechanisms of USP48-mediated regulation of p65 activity in the RPE, with the objective of potentially open new avenues of therapeutic intervention in retinal pathologies by acting on inflammation as a key pathogenic mechanism. We here show that USP48 downregulation stabilizes p65at basal conditions and that nuclear increase in p65 is accompanied by an increased NF-κB transcriptional activity.

## 2. Results

### 2.1. USP48 Overexpression Promotes p65 Nuclear Localization

To study the subcellular localization of USP48 in retinal cells, the human retinal pigment epithelium cells hTERT-RPE1 were transfected with complementary DNA (cDNA) encoding full-length USP48 protein fused to the GFP tag. GFP-USP48 targeted mainly nuclei, which were visualized by cellular staining with DAPI (Figure 1A). However, cytosolic staining was also occasionally seen in individual cells (Figure 1A). We observed that both cytosolic and nuclear GFP-USP48 presented partial colocalization with endogenous p65, suggesting a synergic function in this cell line (Figure 1B,C).

p65 is actively exported from the nucleus, leading to its predominantly cytoplasmic localization in unstimulated cells [7]. TNFα and other inducers of the canonical NF-κB pathway promote degradation of IκBα, exposing the p65 nuclear localization sequence (NLS) and resulting in p65 nuclear localization [29], (Appendix A). However, the p65-mediated transcriptional output is directed not only by the nuclear abundance of p65 but also by several aspects of p65 translocation dynamics [30]. In our experiments we observed that the overexpression of GFP-USP48 correlated with an increased number of unstimulated cells with p65 nuclear localization, suggesting that USP48 could facilitate p65 nuclear translocation. To corroborate this data, we transfected hTERT-RPE1 cells with control GFP, GFP-USP48, GFP-USP48 C98S (a catalytically inactive form of USP48), and GFP-USP25 (not previously related with the NF-κB, thus providing a negative control), and studied the localization of endogenous p65 (Figure 2A). We detected an increased percentage of p65 nuclear localization in GFP-USP48 expressing cells compared with control GFP-transfected cells. This phenotype was specific to wild type GFP-USP48 and was not detected in GFP-USP48 C98S or GFP-USP25 cells, implying this effect might be dependent on USP48 deubiquitinating activity (Figure 2B).

Our findings suggest that USP48 and p65 colocalize in both nuclear and cytosolic compartments. Although colocalization does not imply an interaction between proteins, previous data described a direct interaction between USP48 and p65. Thus, we may postulate that this interaction could facilitate p65 nuclear translocation in resting cells.

### 2.2. USP48 Downregulation Induces Basal p65 Stabilization

To unravel the role of USP48 in modulating p65 activity in retinal cells, we used small interfering RNA (siRNA) to silence *USP48*. To evaluate the knockdown of USP48, we tested a non-targeting siRNA control (scControl) and a siRNA against *USP48* (siUSP48) in both resting and TNFα stimulated hTERT-RPE1 cells by western blot. Densitometric quantification of USP48 protein levels revealed a significant decrease in USP48 expression in both resting and TNFα stimulated cells when treated with siUSP48 (Appendix A). Thus, we chose this tool to silence USP48 in subsequent experiments.

USP48 has been previously described to stabilize nuclear p65 in TNFα-stimulated HeLa cells [17]. To study the effect of USP48 downregulation on p65 protein levels, we knocked down USP48 expression in human hTERT-RPE1 cells by using siUSP48 in both basal conditions and following treatment with TNFα at two different time points: 30 min (early response) and 4 h (late response). Then, we performed immunocytochemistry to quantify the expression levels of p65 in whole cells as well as in nuclear or cytosolic compartments, using p65 integrated density measurements, as stated in the materials and methods section (Figure 3A).

As expected, TNFα treatment induced an increase in the nuclear/total p65 ratio and a decrease in the cytosolic/total p65 ratio in both early and late responses (30 min and 4 h of treatment, respectively; Appendix A). Surprisingly, we detected that in non-treated cells, total p65 significantly increased when USP48 was silenced in comparison with control cells (Figure 3B, NT, left panel). This increase was not restricted to a specific cellular compartment, being detectable in both nuclei and cytosol (Figure 3B, NT, mid and right panels). In contrast, no changes in the localization of p65 in siUSP48- compared to scControl cells were found following TNFα treatment, but for a slight and no statistically significant increase in p65 nuclear levels in UP48 knockdown cells (Figure 3B).

Overall, our data show that downregulation of USP48 levels stabilizes p65 in hTERT-RPE1 at basal conditions.

### 2.3. USP48 Downregulation Increases p65 Nuclear Accumulation

We further extended our data by performing western blot analysis on hTERT-RPE1 cells at both basal conditions (NT) and following treatment with TNFα at different time points (Figure 4A,B). We observed a stabilization of p65 in siUSP48 cells versus scControl cells at basal conditions, while no significant differences were observed following TNFα treatment (Figure 4A). We validated our results in ARPE-19 cells, another human RPE cell line widely used to study retinal biology. To eliminate the possibility that the observed phenotype may be due to off-target effects of the siRNA, we also used a different siRNA against *USP48* (siUSP48#2) that efficiently downregulated *USP48* in ARPE-19 (Appendix A). The stabilization of p65 in siUSP48-trasfected cells was confirmed in ARPE-19 (Appendix A). Finally, we showed that siUSP48#2 efficiently downregulated its target in hTERT-RPE1 and induces p65 stabilization (Appendix A). P65 transcript did not significantly change when USP48 was silenced, suggesting that the changes observed on p65 expression by western blot are very likely due to the regulation by post-translational modifications specifically affecting the p65 protein (Appendix A).

Additionally, p65 stabilization seems to be independent of IκBα, which remains unchanged in siUSP48 cells versus control cells (Figure 4B). Similarly, upstream activation of the NF-κB pathway is not affected, given that P-IκBα/IκBα levels in scControl and siUSP48 cells are similar (Appendix A).

USP48 has been described to regulate the levels of nuclear p65 [17]. We further explored specific changes in p65 expression/accumulation in nuclear or cytosolic compartments by using a more accurate technique than immunocytochemistry. We silenced USP48 in hTERT-RPE1 cells and analyzed the levels of nuclear and cytosolic p65 upon TNFα activation (15 min) by western blot. The nuclear and cytosolic fractions were probed with appropriate controls to ensure that the preparations were free from contaminations. As expected, TNFα induced nuclear translocation of p65, with a decrease in its cytosolic amount. Nuclear levels of p65 in siUSP48 cells were higher than in the control, indicating a nuclear accumulation of p65 in both non-stimulated and TNFα-stimulated cells (Figure 4C). USP48 downregulation stabilizes p65 in the nuclei—where it accumulates following TNFα stimulation– which will thus activate the transcription of target genes. This stabilization appears to be unrelated to IκBα, as the kinetics of IκBα degradation are not affected by USP48 silencing, neither in the nuclear nor cytosolic compartments.

### 2.4. USP48 Downregulation Enhances TNFα-Induced NF-κB Activity

Previous reports have demonstrated that USP48 silencing downregulates TNFα –mediated NF-κB activation in vitro in non-retinal cells [17]. As we observed an increase in nuclear p65 in hTERT-RPE1 cells when USP48 is downregulated, we sought to test also possible changes in the NF-κB transcriptional activity under these conditions. We performed a luciferase-reporter assay by transfecting hTERT-RPE1 cells with either scControl or siUSP48, together with the Ig-kB-Luc plasmid. We observed that USP48 downregulation induced the activation of the NF-κB pathway in both non-stimulated and TNFα-stimulated cells (Figure 5A). The expression levels of USP48 in cellular extracts were assessed by western blot, and TUBULIN was also analyzed, as additional control (Figure 5B). To further confirm our last finding, we checked the expression of well-known targets of NF-κB involved in controlling cell cycle progression: PCNA and Cyclin E. Indeed, the NF-κB pathway positively regulates both PCNA, while transcriptionally represses Cyclin E [31]. We performed western blot analysis on cellular extracts obtained from both siUSP48-transfected cells and control cells, stimulated with TNFα at different time points, and found that USP48 downregulation slightly increased PCNA expression (indicative of an NF-κB positive regulation), and reduced the expression of Cyclin E (indicative of an NF-κB negative regulation), in no TNFα-treated cells. However, these slight differences did not attain statistical significance (Appendix A).

## 3. Discussion

Traditionally, the eye has been considered an immune-privileged site. Contributing to this immune privilege is the barrier formed by the RPE. Besides forming a physical barrier, monolayer RPE cells orchestrate both innate and adaptive immunity and display a plethora of tools to regulate the immune response (e.g., by producing cytokines, activating toll-like receptors, and regulating complement system) [32]. Environmental and genetic factors such as aging, metabolic abnormalities, altered vascular perfusion, or degenerative genetic conditions, may trigger a prolonged and dysregulated immune response, which in turn may contribute to the retinal pathology. Consequently, the refined modulation of the immune response could be crucial to directing the final pathological output in retinal disorders.

In the retina, the pro-inflammatory state is related to the activation of NF-κB, because of an increase in oxidative stress which induces high levels of pro-inflammatory cytokines, such as TNFα and chemokines. Importantly, NF-κB plays a pivotal role in many other retinal processes such as cell death/survival decision, reprogramming of glial cells into neurons (in the rodent retina), and regulation of proliferation. Postransductional modifications exquisitely modulate not only the activation of NF-κB, but also the regulation of NF-κB transcriptional response. NF-κB activation must be terminated at appropriate time points to prevent excessive inflammation. Indeed, in the nuclei, the polyubiquitination of NF-κB leads to its proteasomal degradation and it is a major rate-limiting factor for the expression of proinflammatory genes [33]. Most of the knowledge regarding the ubiquitin-mediated regulation of NF-κB stems from studies on p65. Several ubiquitin E3 ligases targeting p65 for proteasomal degradation have been reported, including SOCS1, COMMD1, PDLIM2, PPAR, and ING4 [33]. Importantly, ubiquitination is reversible: deubiquitinating enzymes (DUBs) can counteract ubiquitination by removing monoubiquitin and polyubiquitin chains bound to the target protein lysine. Deubiquitinases, therefore, can modify a potentially degradative signal and rescue a protein from proteolysis. Deubiquitination can also modify non-degradative signals that are, in turn, regulating signaling cascades. Moreover, some deubiquitinases can modify the polyubiquitin chain length or facilitate the exchange of one type of ubiquitin linkage for another [34].

In humans, there are 102 putative DUB genes, which can be classified into two main classes: cysteine proteases and metalloproteases, which in turn can further divide into genes encoding 58 ubiquitin-specific proteases (USPs), 4 ubiquitin C-terminal hydrolases (UCHs), 5 Machado-Josephin domain proteases (MJDs), 14 ovarian tumor proteases (OTU), and 14 Jab1/Mov34/Mpr1 Pad1 N-terminal+ (MPN+) (JAMM) domain-containing genes. Several DUBs from different families participate in the regulation of NF-κB activation [35]. The first USP protein described to regulate inflammatory response by acting on p65 was the nuclear DUB, USP7. USP7 removes polyubiquitin chains from p65 and promotes transcription by increasing the stability and half-life of transcriptionally active p65 [36]. Subsequently, USP48, belonging to the group of USPs, was also described to regulate NF-κB [16,36].

All the studies aimed to describe the role of USP48 in regulating p65 stability and activity have been performed in non-retinal cells. However, USP48 function in the retina and RPE remains unknown. The relevance of the USP48 function in the retina originally arose from studies performed in the developing retina. *USP48* knockdown in zebrafish resulted in an altered ocular morphology, suggesting that it could be relevant for eye development [37]. Moreover, ChIP-seq data and transcriptomic analysis in mice revealed that the transcription factor CRX (a key regulator of photoreceptor differentiation) binds to the *Usp48* promoter and that *Usp48* is differentially expressed during cone and rod development, being much more expressed in cones [38]. Finally, *USP48* expression has been found altered in pathological eye conditions, such as proliferative diabetic retinopathy [39].

The present study provides a first assessment of USP48 function in modulating p65 activity in hTERT-RPE1 retinal cells. Our results show that downregulation of USP48 does not correlate with alteration of the early steps required for the NF-κB transduction pathway, such as IκBα phosphorylation or IκBα accumulation/degradation. We found that USP48 downregulation is associated with p65 stabilization, in line with an initial study from Tzimas et al. [40], describing that USP48 interacts with p65 and that its overexpression inhibits NF-κB activity downstream of IKKβ in HEK293T cells. On the other hand, Schweitzer K et al., described that upon TNFα-induced p65 nuclear entry, USP48 and CSN associate with p65 antagonizing its ubiquitination and thus preventing the degradation of p65 via proteasome, in HeLa cells [17]. Thus, the mechanism by which p65 stability and activity are regulated by USP48 seems to be highly cell type-dependent.

As it happens for most DUBs, USP48 would be expected to protect its binding partners from degradation due to its inherent deubiquitinase activity. Our findings could be explained by hypothesizing the existence of an intermediary E3 ligase that acts upon p65 and is targeted by USP48. USP48 expression will stabilize this intermediary ligase, and thus p65 will be ubiquitinated and degraded. Upon downregulating USP48, the intermediary ligase will be destabilized, and then, p65 levels would increase (Figure 6). This mechanism of regulation has been already described for USP9X, which downregulates the protein pVHL through its substrate Smurf1, an E3 Ubiquitin ligase that targets pVHL [41]. Further studies will be required to identify the possible intermediary protein linking USP48 activity and p65 stabilization. Importantly, our data indicate that USP48 acts on the nuclear fraction of p65. Consistently, p65 ubiquitination has been described to occur predominantly in the nucleus, and inhibition of proteasome activity appears to selectively stabilize nuclear p65 with minimal effect on the cytoplasmic fraction [42,43]. In fact, p65 binding to DNA is transient and ubiquitination/degradation of the bound fraction ensures a tight time-controlled transcriptional response to TNF-alpha stimuli. However, USP48 may influence transcription of p65 target genes through additional molecular mechanisms, for instance, by modulating the regulative (non-degradative) post-translational modifications on p65, and in turn its association with specific sets of co-activators or co-repressors.

We further tested if *USP48* overexpression could affect the nuclear translocation of p65. Rather unexpectedly, we found that the percentage of cells presenting p65 nuclear localization increased when overexpressing *USP48* compared to control situations. Moreover, this phenotype requires USP48 catalytic activity, given that the USP48 catalytic mutant behaves as the control. This data seems to be in contrast with the silencing findings, in which *USP48* downregulation also correlates with p65 nuclear stabilization. The mechanisms regulating p65 stabilization and activation are very complex and include the exquisite control of cytosolic-nuclear shuttling. We cannot exclude that USP48 could be involved also at this regulatory level, but further studies are required to explore this possibility. However, overexpression studies are not always able to complement the most physiologically relevant data obtained from downregulation studies.

Finally, consistently with the increased nuclear p65 levels, we found an enhanced TNFα-induced NF-κB activity in *USP48* silenced cells, in both unstimulated and TNFα-stimulated cells. To further corroborate this effect by detecting changes in the expression levels of several proteins encoded by NF-κB target genes, we focused on proteins involved in cell cycle regulation (PCNA and Cyclin E), given that hTERT-RPE1 cells are widely used to model cell division, DNA repair, or ciliogenesis, but are less used for cell death studies. Moreover, NF-κB signaling is implicated in the regulation of cell-cycle progression [31,44], and USP48 has been described as a regulator of cell cycle and proliferation [13,15], linked to tumorigenesis [14,45] and also found to be mutated in cancer [46,47], although the molecular mechanisms associating USP48 with NF-κB in the control of cell proliferation have not been clearly described yet. However, no significant statistical differences were found in PCNA and Cyclin E levels between control and siUSP48-transfected cells, indicating that proliferation might not represent the main process regulated by USP48-mediated NF-κB regulation.

Altogether our data posit USP48 as an important regulator of NF-κB activity in the retina, with mechanisms of intervention on both basal and TNFα-induced NF-κB pathways, which are specific to RPE cells and may also extend to other retinal cells (Figure 6). This finding becomes especially relevant due to the role of the RPE in controlling the retinal immune response. In the last decades several drugs that neutralize TNFα—i.e., TNFα blockers—have been developed to reduce symptoms of various diseases in which TNFα acts as a pathogenic driver [48,49,50,51]. Indeed, understanding the immune regulatory properties of the RPE may provide additional clues to understanding disease mechanisms and open new paths to develop effective therapeutical strategies for many human retinal diseases, including both rare inherited retinal dystrophies as well as high prevalence disorders associated with retinal degeneration and inflammation, such as age-related macular degeneration or diabetic retinopathy.

## 4. Materials and Methods

### 4.1. Cell Culture and Transfections

Human hTERT-RPE1 cells were cultured in 10% fetal bovine serum (FBS) and 1% penicillin/streptomycin in 1:1 Dulbecco’s Modified Eagle’s Medium (DMEM) (ATCC, Manassas, VA, USA) and Ham’s F-12 Nutrient Mix (F12) (Life Technologies, Carlsbad, CA, USA) in a 5% CO_2_ cell culture humidified incubator at 37 °C. Plasmid transfection was performed using Lipofectamine 3000 (Invitrogen) following the manufacturer’s instructions and processed 24 h after transfection. siRNA transfection was performed by transfecting cells in suspension with lipofectamine RNAiMAX reagent (Thermo Fisher Scientific, Rockford, IL, USA) and 10 nM of either scrambled siRNA (scControl) (Dharmacon, D-001810-01-05) or anti-USP48 small interfering RNA (siUSP48) (Dharmacon, J-006079-11-0002) and anti-USP48 small interfering RNA (siUSP48#2) (Dharmacon, J-006079-09-0002). Cells were also processed 24 h after transfection. In immunocytochemistry experiments, hTERT-RPE1 cells were seeded on coverslips in 24-well plates. Mouse Tumor Necrosis Factor-α (TNFα) was purchased from Roche (Basel, Switzerland, cat. number: 11271156001) and used with a final concentration of 20 ng/mL.

### 4.2. Plasmid Vectors

pcDNA6.2-emGFP-USP48 and pcDNA6.2-emGFP-USP48 C98S expression constructs were generous gifts from Dr. Joanna I. Loizou [11], and Ig-kB-luc was a generous gift from Dr. Matilde Valeria Ursini [52]. pEGFP and pEGFP-USP25 constructs were previously described [53].

### 4.3. Immunofluorescence

In immunocytochemistry experiments, cells were fixed in pre-chilled methanol at −20 °C for 10 min, washed in PBS 1× (3 × 5 min), permeabilized in 0.2% Triton X-100 (Sigma-Aldrich, St. Louis, MO, USA) in PBS (20 min at RT), and blocked for 1 h in 10% Normal Goat Serum (Roche Diagnostics, Indianapolis, IN, USA) in PBS. Primary antibodies were incubated overnight at 4 °C in a blocking solution. The primary antibodies used were: p65 (Cell Signaling Technology, Danvers, MA, USA; 8242, 1:500) and TUBULIN (Sigma-Aldrich, Saint Louis, MO, USA; T5168, 1:500). After incubation, coverslips with cells were rinsed in PBS 1× (3 × 5 min), incubated with the corresponding secondary antibodies (AlexaFluor 488 anti-Mouse) (Thermo Fisher Scientific, Rockford, IL, USA; A11017; 1:500), AlexaFluor 488 anti-Rabbit (Thermo Fisher Scientific, Rockford, IL, USA; A11070; 1:500) at RT (1 h) in blocking solution. Nuclei were stained with DAPI (Roche Diagnostics, Indianapolis, IN, USA) (1:1000), washed again in PBS 1× (3 × 5 min), and mounted in Mowiol 4–88 (Merck, Darmstadt, Germany).

### 4.4. Microscope Image Acquisition

Samples were analyzed by confocal microscopy (Zeiss LSM 880, Thornwood, NY, USA) and images were collected using ZEN-LSM software. For quantitative analysis of the percentage of GFP positive cells with p65 nuclear localization, the number of GFP positive cells expressing p65 inside the nucleus was counted manually with ImageJ software (National Institutes of Health, Bethesda, MD). The analysis was performed in 11–17 fields per condition, from 2 independent experiments.

For quantitative analysis of p65 in the cytosolic or nuclear compartment, we used DAPI and TUBULIN stainings to specifically delimitate the cytosolic and nuclear compartments, respectively. Integrated density from p65 (red) channel was measured by using ImageJ software (National Institutes of Health, Bethesda, MD, USA) and values were normalized against the scControl-transfected cells (non-treated). The analysis was performed in 45 cells per condition from 3 independent experiments.

### 4.5. Western Blot

Cells were lysed in SDS-PAGE protein loading buffer 1× (60 mM TrisHCl (pH 6.8), 10% glycerol, 2% SDS and 0.1% bromophenol blue) or assay buffer (50 mM Tris pH 7.4, 150 mM NaCl, 1 mM EDTA, 1% NP-40, 0.25% Na-deoxycholate, and protease inhibitors). Proteins were analyzed by SDS-PAGE and transferred onto PVDF membranes, which were blocked with 3% BSA in PBS-buffered saline (PBS) containing 0.1% Tween 20 and incubated overnight at 4 °C with primary antibodies. After incubation with horseradish peroxidase-labeled secondary antibodies for 1 h at room temperature, membranes were revealed with the ECL system (Lumi-Light Western Blotting Substrate, Roche). Images were acquired by ImageQuant™ LAS 4000 mini Image Analyzer (Fujifilm, Tokyo, Japan) and quantified using ImageJ software. TUBULIN loading control was used when required. The primary antibodies used were the following: USP48 (Abcam, Cambridge, UK; ab72226, 1:1000), p65 (Cell Signaling Technology, Danvers, MA, USA; 8242, 1:1000), IκBα (Cell Signaling Technology, Danvers, MA, USA; 4814, 1:1000), P-IκBα (Cell Signaling Technology, Danvers, MA, USA; 2859, 1:1000), Cyclin E (Abcam, Cambridge, UK; ab2094; 1:1000), PCNA (Abcam, Cambridge, UK; ab29; 1:1000), NUCLEOLIN (Abcam, Cambridge, UK; ab70493; 1:1000); TUBULIN (Sigma-Aldrich, St. Louis, MO, USA; T5168, 1:1000).

### 4.6. Nuclear/Cytosol Subcellular Fractionation

Cells grown on 6-well plates were washed twice with PBS, harvested in cold PBS, and processed using the Nuclear/Cytosol Fractionation Kit (Abcam, ab289882), following the manufacturer’s instructions.

### 4.7. Luciferase Assay

hTERT-RPE1 cells were transfected with 10 nM scControl or siUSP48 and the reporter plasmid (Ig-kBluc). Twenty-four hours after transfection, the medium was refreshed and the cells were stimulated with TNFα (20 ng/mL). Four hours later, the cells were lysed with PLB 1× (Passive Lysis Buffer, Promega, Madison, WI, USA) and the luciferase activity of the reporter expression construct Ig-kB-luc was assessed using the Luciferase Assay system (Promega, Madison, WI, USA), by following the manufacturer instructions.

### 4.8. Real-Time-PCR

RNA was extracted from hTERT-RPE1 cells cultured on 6-well plates using the RNeasy Plus Mini Kit (QIAGEN, Hilden, Germany; 74134) according to the manufacturer’s instructions. Samples were incubated for 1 h at 37 °C with 2 units of AmbionTM DNase I (Thermo Fisher Scientific, Rockford, IL, USA; AM2222). Next, samples were further purified with the High Pure RNA Isolation Kit (Roche Diagnostics, Indianapolis, IN, USA; 11828665001). RNA was reverse transcribed using the qScript cDNA Synthesis Kit (QuantaBio, Beverly, MA, USA; 95047-100) following the manufacturer’s instructions. Specific primers for amplifications were designed and optimized (hP65_FW:CTACGACCTGAATGCTGTGC; hP65_RV:CTGCCAGAGTTTCGGTTCAC; hTNFAIP3_FW:CTGAAATCCGAGCTGTTCCAC; hTNFAIP3_RV:GAGATGAGTTGTGCCATGGTC; hACTB_FW:CATCCGCAAAGACCTGTACG; hACTB_RV: CCTGCTTGCTGATCCACAT). Real-time PCR (qPCR) was performed according to standard thermocycling conditions using the LightCycler 480^®^ SYBR Green I Master (Roche Diagnostics, Indianapolis, IN, USA; 04707516001) and a LightCycler^®^ 480 Multiwell Plate 384 (Roche Diagnostics GmbH, Penzberg, Germany) in a final volume of 10 µL. Raw data were analyzed with the LightCycler^®^ 480 software and gene expression levels were determined using the delta-delta Ct method: *ACTB* expression was employed as the housekeeping or normalizer gene and the unstimulated scControl condition was the untreated sample.

### 4.9. Statistical Analyses

Data were analyzed using GraphPad Prism software (GraphPad6 Software Inc., San Diego, CA, USA). Homoscedasticity and normality were verified using Bartlett’s test, and D’Agostino and Person’s test, respectively. When data were homoscedastic and followed a normal distribution, they were analyzed by t-test, one-way ANOVA, and two-way ANOVA. Kruskal–Wallis tests were used when data did not follow a normal distribution.

## Figures and Tables

**Figure 1 ijms-23-09682-f001:**
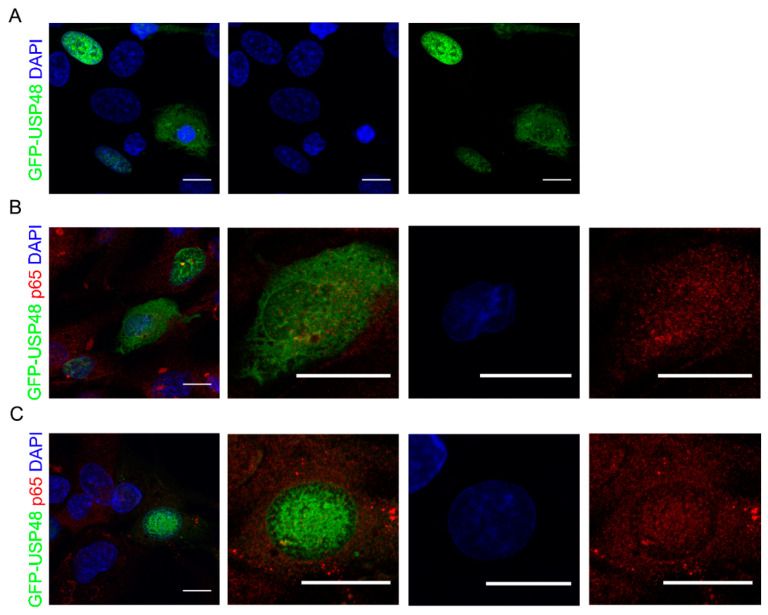
USP48 colocalizes with p65 and its overexpression promotes p65 nuclear localization. Immunofluorescence images from hTERT-RPE1 transfected with GFP-USP48 (green). (**A**) GFP-USP48 is localized mainly in the nuclei, while cytosolic staining is occasionally seen in individual cells. (**B**,**C**) Endogenous p65 (red) partially colocalizes with GFP-USP48 in both cytosolic and nuclear compartments. Regions of interest (ROIs) represent higher magnification from left panels. Nuclei were stained with DAPI (blue). Scale bars: 15 μm and 20 μm in higher magnification images.

**Figure 2 ijms-23-09682-f002:**
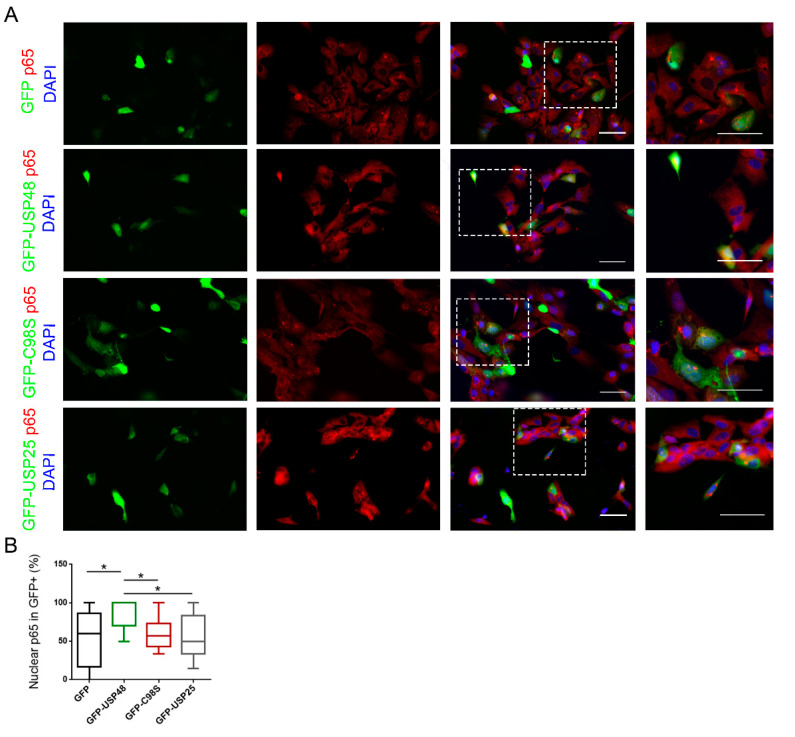
USP48 overexpression facilitates p65 nuclear localization. (**A**) Immunofluorescence images from hTERT-RPE1 transfected with either GFP (empty vector), GFP-USP48, GFP-USP48 C98S (catalytically inactive mutant), or unrelated GFP-USP25 (green), and immunostained with p65 (red). Nuclei were counterstained with DAPI (blue). Scale bar: 50 μm in both lower and higher magnification images. (**B**) Quantification of the percentage of transfected cells presenting nuclear p65. The percentage of GFP-USP48 (WT) expressing cells showing p65 nuclear localization is increased in comparison with either GFP, GFP-USP48 C98S, or GFP-USP25. Box plots (min to max) were obtained from 11–17 images per condition from 2–3 independent replicates. Statistical analysis was performed by one-way ANOVA: * *p*-value ≥ 0.05.

**Figure 3 ijms-23-09682-f003:**
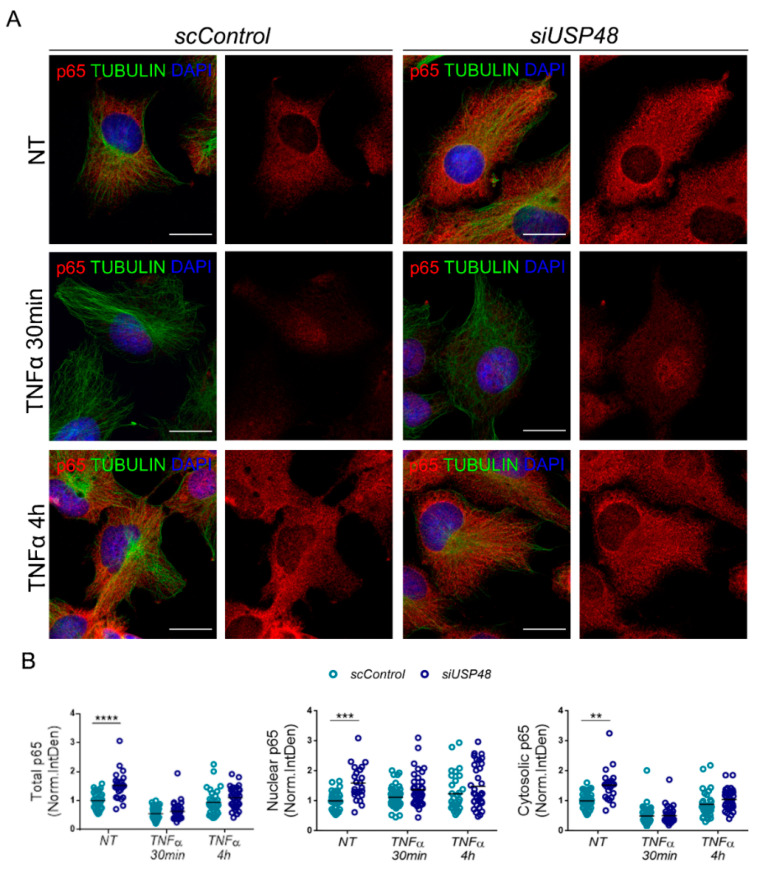
USP48 downregulation stabilizes p65. (**A**) Immunocytochemistry of anti-p65 (red) in USP48 knockdown hTERT-RPE1 cells at basal conditions and after treatment with TNFα at two different time points: 30 min (early response) and 4 h (late response). Cells were counterstained with anti-tubulin (green) to highlight the cell shape, and DAPI to delimit the nucleus (blue). (**B**) Total p65 (Integrated density) was significantly increased in USP48 silenced cells (*siUSP48*, dark blue dots) in comparison with control cells (scControl, light blue dots), being this increase was detectable in both nuclear and cytosolic compartments. No significant changes were found following TNFα treatment at 30 min or 4 h. Scale bar: 20 μm. Data are represented as the mean, n = 33–45 cells per condition from 3 independent experiments. Statistical analysis by two-way ANOVA: ** *p*-value ≥ 0.01; *** *p*-value ≥ 0.001; **** *p*-value ≥ 0.0001.

**Figure 4 ijms-23-09682-f004:**
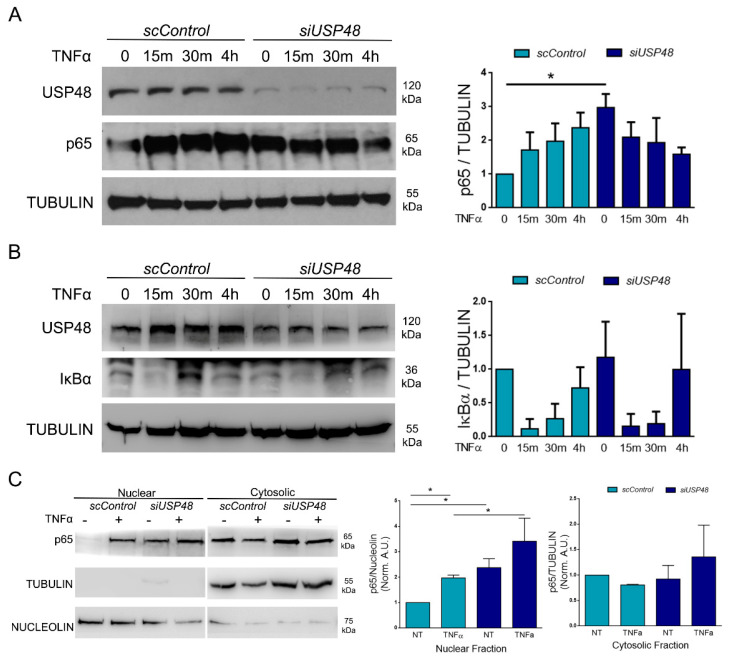
USP48 downregulation stabilizes p65, mainly in the nuclear compartment. USP48 expression was downregulated in hTERT-RPE1 cells at both basal conditions (NT) and after treatment with TNFα at several different time points. Cells were harvested and processed for immunoblotting. (**A**) Densitometric quantification indicates that p65 levels significantly increased in NT cells. (**B**) No changes were detected in IκBα expression levels. (**C**) Nuclear and cytosolic extracts were prepared for immunoblotting with anti-USP48 and p65 antibodies. Anti-nucleolin (nuclear) and anti-tubulin (cytosolic) were used to detect both enrichment and contamination of fractionated cell samples. p65 levels increased specifically in the nuclear compartment when USP48 is downregulated. Images represent one of three (**A**,**C**) or four (**B**) independent experiments. Statistical analysis by Kruskal Wallis in (**A**,**B**): * *p*-value ≥ 0.05 and by Student’s *t*-test in (**C**): * *p*-value ≥ 0.05.

**Figure 5 ijms-23-09682-f005:**
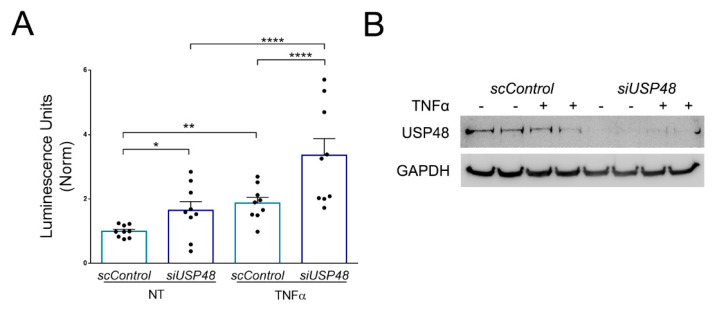
*USP48* silencing enhances NF-κB activity. (**A**) hTERT-RPE1 cells were transfected with scControl or siUSP48, in combination with a 3-kB-luciferase reporter plasmid (Igk-BLuc). Twenty-four hours after transfection, cells were stimulated for 4 h with TNFα and then lysed to measure NF-κB activity. n = 9 from three independent experiments conducted in triplicate. Data are represented for each experiment as normalized mean ± S.E. Statistical analysis by two-way ANOVA: * *p*-value ≥ 0.05; ** *p*-value ≥ 0.01; **** *p*-value ≥ 0.0001. (**B**) USP48 expression was efficiently silenced by siUSP48 (representative western blot). Tubulin was used as a loading control.

**Figure 6 ijms-23-09682-f006:**
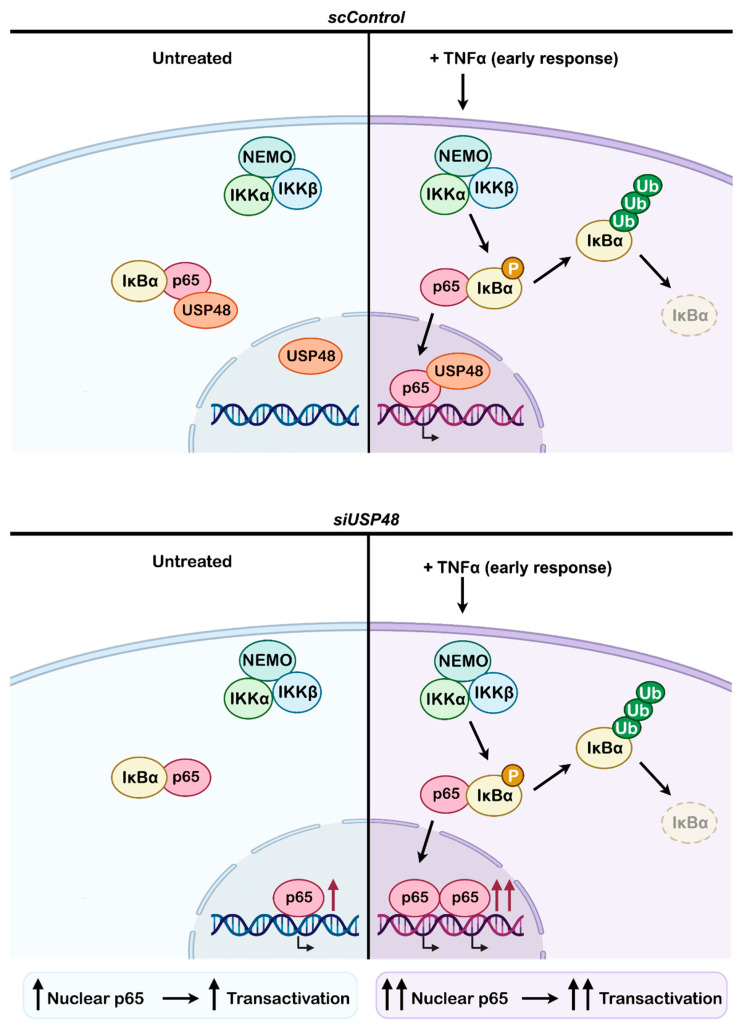
USP48 downregulation induces the activation of the NF-κB pathway. In control (scControl) untreated conditions, p65 is sequestered in the cytoplasm by IκBα. Upon TNFα stimulation, IκBα is phosphorylated by the IKKα:IKKβ:NEMO complex, resulting in IκBα ubiquitination and subsequent proteasomal degradation. As a consequence, the liberated p65 translocates to the nucleus, where it activates the transcription of target genes and can interact with USP48. When USP48 is downregulated (siUSP48), p65 levels increase –mainly in the nucleus– inducing the transcriptional activation of NF-κB target genes, even without TNFα stimulation. The levels of p65 further increase in the nucleus upon TNFα treatment, in an additive response, thus resulting in further transactivation of target genes.

## Data Availability

The data presented in this study are available in this article and Appendix A.

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
