# Peer review of "Ubiquitin Specific Protease USP48 Destabilizes NF-κB/p65 in Retinal Pigment Epithelium Cells"

_ijms, 2022, doi:10.3390/ijms23179682_

Round 1

Reviewer 1 Report

The manuscript entitled “Ubiquitin specific protease USP48 destabilizes NFkB/p65 in retinal pigment epithelium cells” by Mirra et al. provides evidence that knocking down the deubiquitinating enzyme USP48 in a human retinal pigment epithelium cell line (hTERT-RPE1) leads to increased protein level of p65, a key factor of the NFkB pathway. Understanding the factors that modulate the NFkB pathway in retinal cells is important, as excessive inflammatory responses have been associated with many retinal neurodegenerative diseases. Although this study delivers potentially useful information regarding the unknown role of USP48 in retinal cells, the main conclusions of the manuscript lack experimental support.

Most importantly, the main phenotype of the study (i.e. cell-wide increase in p65 protein level) is observed from knocking down USP48 using one siRNA in one cell line. At least two siRNAs would need to be used for these experiments to reduce confounding off-target effects. Ideally, KO cell lines need to be created to demonstrate the increase in p65 level upon depleting USP48. And this phenotype should be rescued by compensating the KO cells with exogenously expressed USP48. The current design of the experiments (using one single siRNA) does not eliminate the possibility that the phenotype observed may be due to off-target effects of the siRNA. Additionally, the main findings should be validated in another RFE cell line to make any general conclusions on the role of USP48 in retina.

The authors claim that USP48 downregulation stabilizes p65. While this conclusion is drawn from comparing the protein levels of p65 (using immunofluorescent microscopy and western blot) in siControl vs. siUSP48 cells, no experiments have been done to compare the ubiquitination levels or the half-lives of p65. Therefore, the data presented in the manuscript is insufficient to show that the increased p65 level in the siUSP48 cells is due to protein stabilization.

A few more specific comments:

Figure 2A is not very clear at showing the localization of p65. Can the authors provide images with higher magnification as well as DAPI staining? How is the quantification of % nuclear p65 performed without DAPI?

Due to uneven loading of the nuclear fractions, Figure 4C is not clear enough to demonstrate that USP48 KD specifically causes nuclear accumulation of p65. It would be helpful if the authors can quantify the bands, normalize them to loading controls (nucleolin and tubulin), and compare the ratio of nuclear-to-cytosolic p65 in different conditions.

With Figure 5C, the authors claim that USP48 KD increases both Cyclin D1 and PCNA levels. The phenotypes seem very subtle on the blots. Can the authors quantify the bands and show statistical significance of the difference?

The references seem to be irrelevant or incorrect in at least two places - Ref #11 at line 61 and Ref #38 at line 311. 

Author Response

ANSWERS TO REVIEWER 1

The manuscript entitled “Ubiquitin specific protease USP48 destabilizes NFkB/p65 in retinal pigment epithelium cells” by Mirra et al. provides evidence that knocking down the deubiquitinating enzyme USP48 in a human retinal pigment epithelium cell line (hTERT-RPE1) leads to increased protein level of p65, a key factor of the NF-kB pathway. Understanding the factors that modulate the NF-kB pathway in retinal cells is important, as excessive inflammatory responses have been associated with many retinal neurodegenerative diseases. Although this study delivers potentially useful information regarding the unknown role of USP48 in retinal cells, the main conclusions of the manuscript lack experimental support.

Most importantly, the main phenotype of the study (i.e. cell-wide increase in p65 protein level) is observed from knocking down USP48 using one siRNA in one cell line. At least two siRNAs would need to be used for these experiments to reduce confounding off-target effects. Ideally, KO cell lines need to be created to demonstrate the increase in p65 level upon depleting USP48. And this phenotype should be rescued by compensating the KO cells with exogenously expressed USP48. The current design of the experiments (using one single siRNA) does not eliminate the possibility that the phenotype observed may be due to off-target effects of the siRNA. Additionally, the main findings should be validated in another RFE cell line to make any general conclusions on the role of USP48 in retina.

We thank the Reviewer for his/her comment. We totally agree about the importance of creating USP48-KO cell lines to demonstrate the increase in p65 levels upon depleting USP48 and perform experiments to rescue the molecular phenotype. However, due to the limited time available, we took different approaches to strengthen our conclusions, as kindly suggested by the Reviewer. In the revised version of the manuscript, we have added further validations of our results in the ARPE-19 cell line, a human RPE cell line widely used to study retinal biology and included a new Supplementary Figure (Figure S4). In these cells, we have compared the expression levels of USP48 after silencing with two different siRNAs against USP48 (namely, siUSP48 and siUSP48#2) to the controls of non-transfected cells as well as cells transfected with two different scrambled siRNAs (scControl -originally used in the first version of the manuscript)- and scControl#2. We demonstrated that both siUSP48 and siUSP48#2 efficiently downregulated USP48 in ARPE-19 (new Figure S4A). Moreover, the stabilization of p65 in siUSP48-trasfected cells was confirmed in ARPE-19 (new Figure S4B). Finally, to avoid the possibility of confounding off-target effects, we showed that siUSP48#2 efficiently downregulated its target in hTERT-RPE1 and induces p65 stabilization (new Figure S4C).

The authors claim that USP48 downregulation stabilizes p65. While this conclusion is drawn from comparing the protein levels of p65 (using immunofluorescent microscopy and western blot) in siControl vs. siUSP48 cells, no experiments have been done to compare the ubiquitination levels or the half-lives of p65. Therefore, the data presented in the manuscript is insufficient to show that the increased p65 level in the siUSP48 cells is due to protein stabilization.

We agree with the Reviewer about the importance to corroborate that the increase in p65 levels in siUSP48-trasfected cells is due to protein stabilization. Unfortunately, we were not able to detect clear changes in the ubiquitination levels of p65 neither by western blot nor by immunoprecipitating p65 and performing western blot against overexpressed tagged-ubiquitin. To discard whether the lower protein levels of p65 were due to transcriptional mechanisms, we performed RT-PCR experiments and showed that the levels of P65 transcript did not significantly change when USP48 was silenced. Therefore, the changes observed on p65 expression by western blot are very likely due to the regulation by post-translational modifications specifically affecting the p65 protein. Note that the NF-kB target gene TNFAIP3 was used as positive control (new Figure S4D).

A few more specific comments:

Figure 2A is not very clear at showing the localization of p65. Can the authors provide images with higher magnification as well as DAPI staining? How is the quantification of % nuclear p65 performed without DAPI?

In this new version, we modified Figure 2 according to the Reviewer suggestions. The images are now with higher magnification for the three channels used, including DAPI, that was left out in the first version of the manuscript.

 Due to uneven loading of the nuclear fractions, Figure 4C is not clear enough to demonstrate that USP48 KD specifically causes nuclear accumulation of p65. It would be helpful if the authors can quantify the bands, normalize them to loading controls (nucleolin and tubulin), and compare the ratio of nuclear-to-cytosolic p65 in different conditions.

As suggested by the referee, we modified Figure 4C, including images from a more representative western blot. Moreover, we quantified the bands obtained for both non-treated and TNFa-treated cells (15 minutes of treatment), normalizing them to loading controls (nucleolin and tubulin) to confirm that USP48 silencing specifically induces nuclear accumulation of p65.

With Figure 5C, the authors claim that USP48 KD increases both Cyclin D1 and PCNA levels. The phenotypes seem very subtle on the blots. Can the authors quantify the bands and show statistical significance of the difference?

We followed the reviewer suggestions and quantified the bands obtained in blots of n=3-4 independent experiments. As the obtained data did not follow a normal distribution, statistical analysis was performed by non-parametric Kruskal Wallis test, which did not show statistical significance between conditions. However, at least for Cyclin E and PCNA, we can appreciate a subtle phenotype. We present Cyclin E and PCNA data (western blots and quantifications) in a new Supplementary figure (Figure S6) in the revised manuscript. Moreover, we have added some sentences in the Discussion about proliferation not representing the main process regulated by USP48-mediated NF-κB regulation (now in lines 375-378 in the Discussion section).

The references seem to be irrelevant or incorrect in at least two places - Ref #11 at line 61 and Ref #38 at line 311.

We apologize for the mistake related to the Ref#11, which has been corrected. Ref#38 is correct (Tse et al., BMC Genomics 2009, 10, doi:10.1186/1471-2164-10-637): the effects of usp48 knockdown on eye development in zebrafish can be found embedded in the Table presented in the Additional file 4 of the referenced article.

Reviewer 2 Report

In this manuscript Mirra and colleagues described the effect of down regulation of USP48 on p65 in retinal pigment epithelium in basal conditions and stimulated by TNFalpha. The main objective is to investigate the rol of USP48 in the regulation of NF-kB pathway and its role in inflammatory response. The work is well conceived and contributes to the understanding of the objectives, although the originality is restricted to this type of cells. In this confining context, the manuscript well written and the results are significant, although some concerns rise from its lecture:

To follow better the line of reasoning in the manuscript, the role of USP48 as a DUB should be stressed.

In lines 147 to 149, colocalization does not imply interaction between the proteins, this should be stressed in this sentence.

In line 150, In the title should be add "basal" stabilization, in relation with the sentence in line 185, the "slight increase in p65 nuclear levels" it is not clear for me.  Thus, in lines 186-187, I should eliminate the term "mainly" in this conclusions.

In Fig 4A it is not clear which comparison is significant. Fig 4 B does not show statistical comparisons and in Fig 4C lacks the quantifications of the data.

Author Response

ANSWERS TO REVIEWER 2

In this manuscript Mirra and colleagues described the effect of down regulation of USP48 on p65 in retinal pigment epithelium in basal conditions and stimulated by TNFalpha. The main objective is to investigate the rol of USP48 in the regulation of NF-kB pathway and its role in inflammatory response. The work is well conceived and contributes to the understanding of the objectives, although the originality is restricted to this type of cells. In this confining context, the manuscript well written and the results are significant, although some concerns rise from its lecture:

To follow better the line of reasoning in the manuscript, the role of USP48 as a DUB should be stressed.

We thank the Reviewer for the suggestion. In this revised version of the manuscript we briefly described the role of USP48 as a DUB in the Introduction section (lines 72-77).

In lines 147 to 149, colocalization does not imply interaction between the proteins, this should be stressed in this sentence.

As suggested, in the revised version of the manuscript we have stressed the notion by which colocalization does not imply interaction between proteins (lines 155-157).

In line 150, In the title should be add "basal" stabilization, in relation with the sentence in line 185, the "slight increase in p65 nuclear levels" it is not clear for me.  Thus, in lines 186-187, I should eliminate the term "mainly" in this conclusions.

Thank you for your valuable comment. We corrected the mentioned title, now on line 172 following the Reviewer’s suggestion and eliminated "mainly" from the text (modified sentence now in line 197-198).

In Fig 4A it is not clear which comparison is significant. Fig 4 B does not show statistical comparisons and in Fig 4C lacks the quantifications of the data.

As the data presented in both Figures 4A and 4B did not follow a normal distribution, statistical analysis was performed by non-parametric Kruskal Wallis test. We focused on the comparisons between scControl cells and siUSP48-transfected cells. In Figure 4A, the only significant comparison was found in non-treated cells (scControl cells versus siUSP48-transfected cells, at time=0 of TNFa), whereas no significant comparisons were found in Figure 4B.

As suggested by both the referees, we modified Figure 4C, including images from a more representative western blot. Moreover, we quantified the bands obtained for both non-treated and TNFa-treated cells (15 minutes of treatment), normalizing them to loading controls (nucleolin and tubulin) to confirm that USP48 silencing specifically induces nuclear accumulation of p65.

Round 2

Reviewer 1 Report

The authors have addressed my comments and concerns.